# Digital Magnetic Compass Integration with Stationary, Land-Based Electro-Optical Multi-Sensor Surveillance System

**DOI:** 10.3390/s19194331

**Published:** 2019-10-07

**Authors:** Branko Livada, Saša Vujić, Dragan Radić, Tomislav Unkašević, Zoran Banjac

**Affiliations:** Vlatacom Institute, Milutina Milankovića 5, 11070 Belgrade, Serbia

**Keywords:** electro-optical surveillance system, digital magnetic compass, compass integration, hard iron compensation, soft iron compensation, magnetic heading angle, swinging method, measurement accuracy

## Abstract

Multi-sensor imaging systems using the global navigation satellite system (GNSS) and digital magnetic compass (DMC) for geo-referencing have an important role and wide application in long-range surveillance systems. To achieve the required system heading accuracy, the specific magnetic compass calibration and compensation procedures, which highly depend on the application conditions, should be applied. The DMC compensation technique suitable for the operation environment is described and different technical solutions are studied. The application of the swinging procedure was shown as a good solution for DMC compensation in a given application. The selected DMC was built into a system to be experimentally evaluated, both under laboratory and field conditions. The implementation of the compensation procedure and magnetic sensor integration in systems is described. The heading accuracy measurement results show that DMC could be successfully integrated and used in long-range surveillance systems providing required geo-referencing data.

## 1. Introduction

The multi-sensor imaging systems have a very important role and wide applications in long-range surveillance and security systems [1,2,3,4,5]. They are designed as the adaptable modular system with the capability of managing sensors mounted at the sensor head and fusion of the sensor data to provide data regarding objects of interest (target) inside a field of regard (FOR). All sensor data are collected, analyzed, and presented to the human operator on the operator’s console. Because of that, the structure and functions incorporated in the operator’s console are usually complex and demanding. One of the tasks, in some applications, is to provide target geo-referenced data to be presented on the screen or map. That task is usually achieved using a laser range finder (LRF), a global navigation satellite system (GNSS) receiver, and a heading sensor (north finding) [6].

There are several ways to provide heading data [7]. Most navigation systems today use some type of digital magnetic compass (DMC) to determine heading direction. Using the Earth’s magnetic field component measurements, electronic compasses based on magneto-resistive (MR) sensors can electrically resolve better than 0.1 degree heading change which is good enough for the application in long-range surveillance systems. It is a technically complicated task due to the nature of the Earth’s magnetic field measurements. Specific calibration and compensation techniques should be applied to correct for compass tilt angles and nearby ferrous material disturbances.

The application of the magnetic compass has a several-centuries-long application history [8,9] for ship navigation purposes. Most navigation systems developed today use some type of compass to determine heading direction [10,11] in modern navigation systems, but the technical solution developed depends highly on the application nature.

The basic principles of magnetism and magnetic field measurements were reviewed [12,13], showing that magnetic field measurements and magnetometers could find wide application for building magnetic sensors [14]. Using MEMS (micro electro-mechanical systems) technology for magnetic sensor manufacturing [15,16,17] provides opportunities to make low-cost digital magnetometers and accordingly digital magnetic compasses for the application in navigation systems [18], military systems [19], directional drillers’ geo-referencing systems [20], space navigation systems [21], and automotive systems [22].

In this paper we explore the application of the DMC technology in the long-range multi-sensor electro-optical system to provide support for selected target geo-referencing. We used a DMC from the market. The manufacturer’s magnetometer calibration procedure was provided as a part of the development kit. This calibration procedure provided corrections to support the required heading angle accuracy measurements. Additional heading angle compensation was necessary to provide hard and soft iron compensation on the site of system application. The DMC experimental investigation, compensation procedure selection, and implementation are described. Experimental results are presented that show selected DMC suitability for a given application.

The practical integration of the DMC into an electro-optical multi-sensor system providing the required heading accuracy and using a simplified compensation procedure is described. The system architecture, mission, and components properties allow simplification and system orientation redundancy for system and target geo-referencing. The integration process was finished, and the explored technical solution was successfully applied in the production of long-range surveillance systems. During the application, the system should be calibrated and compensated any time it is positioned at the desired location. DMC calibration and compensation could be done during the whole system activation time.

## 2. Digital Magnetic Compass (DMC) Compensation and Integration

The multi-sensor surveillance system is an adaptable modular system for managing sensors mounted at a sensor head using a human-observed command and control station. The design of the multi-sensor surveillance system depends on many multidisciplinary fields like imaging sensor technology, image processing, position sensing technology, motion control, communication, and networking technology. The selection of the surveillance system components and defining the system architecture is a complex task. There is no universal solution for all situations, but only optimization for aimed application. In the case when a selected target geo-location is required as the system’s output, a digital magnetic sensor could be added in connection with a GNSS receiver and an eye-safe laser rangefinder to provide the required data.

### 2.1. Long-Range Electro-Optical (EO) Surveillance System

The generalized multi-sensor system structure and architecture providing required functionalities is shown in Figure 1 [23].

The multi-sensor system architecture is comprised of the following components:

**Imaging Group**—provides images of the object and space of interest (target). Imaging sensors should have the capability of selecting the field of view (FOV), focus control, and calibrations. System could be composed using any of the items listed below, or a combination of such:Day light (low light) video camera,Infra Red—IR Imager (Short Wavelength IR—SWIR, Medium Wavelength IR—MWIR, Long Wavelength IR—LWIR).

**Position Sensing Group**—provides position-related data using:DMC (digital magnetic compass)—north-finding sensor, used for the system’s initial orientation and azimuth measurements requiring heading reading accuracy to provide the required target Circular Error Probability 50% - CEP50 error [24].GNSS receiver—determines system location using data from global navigation satellite systemsLRF (laser range finder)—used for selected target distance measurements.Pan/tilt platform—provides sensor-head-controlled motion, aiming, and position data, having high accuracy at the same time.

Imaging sensor video signals and position sensor data are collected using the computer’s built-in operator’s console and the integrated using system command and control software package. An advanced solution for image enhancement and sensor data fusion could be optionally added to provide the required data on the operator’s console display.

### 2.2. Digital Magnetic Compass—DMC Technology

The measurement of the Earth’s magnetic field direction is one of the oldest and most reliable ways for direction finding and orientation [7]. Traditionally, visual tracking of the magnetized needle motion and its position when aligned with the Earth’s magnetic field, was the way to find the magnetic north direction, and this data was used for orientation. The development of electronic magnetometers that transfer the magnetic field into an electrical signal, using MEMS technology, allows to build an electronic magnetic compass [15,16,17] that comprises three orthogonally mounted magnetometers used for the Earth’s magnetic field component measurements (see Figure 2a). In the case that magnetometers have analog-to-digital converters built in to digitalize the magnetometer’s electrical signal, we are talking about a digital magnetic compass (DMC). In the case when the electronic compass is used in connection with gimbaled systems, the two additional vertical declination IMU (inertial measurement unit) sensors providing compass roll and pitch angle data are incorporated into the DMC. Roll and pitch angle data are used in the DMC for heading angle calculation but can also be reported separately (see Figure 2b).

Using magnetometer measurements values (*B_x_, B_y_, B_z_* in the sensor reference frame), their projected values in the horizontal plane ($B_{x}^{h},\;B_{y}^{h}$ ) can be calculated as per Equation (1), and the heading (magnetic north) angle, *ψ*, can be calculated using Equation (2).
(1)$$\left[\begin{array}{c} B_{x}^{h}\\ B_{y}^{h} \end{array}\right]=\left[\begin{array}{ccc} cos\varphi & sin\theta sin\varphi & -cos\theta sin\varphi \\ 0 & cos\theta & sin\theta \end{array}\right]\left[\begin{array}{c} B_{x}\\ B_{y}\\ B_{z} \end{array}\right],$$
(2)$$\psi =-\tan^{-1}\frac{B_{y}^{h}}{B_{x}^{h}}$$
where *θ* is compass pitch (elevation), and *ϕ* is compass roll (bank) angle against horizontal plane, as illustrated in Figure 2b. DMC provides the compass heading angle output according to Equations (1) and (2) and measured pitch and roll angles.

DMC manufacturers usually provide access to all necessary values including calculated heading angle. In addition, to provide better accuracy, manufacturers provide a factory-designed calibration procedure for magnetometer sensitivity calibration, internal error compensation, and equalization through measurement channels.

DMC is a relatively cheap, compact, lightweight, reliable, and accurate (up to 0.25° (1σ error)) direction finding sensor that could be successfully integrated with other sensors, but carefully applied procedures for compass deviation according to local magnetic influences should be applied.

In our system, we used DMC with basic parameters as listed in Table 1. The DMC is designed to provide 3D magnetic field vector digital values, the azimuth angle, and elevation/bank angles. The DMC development kit provided by the manufacturer also has a magnetometer proprietary software controlled calibration procedure that could be suited to the application. After calibration, DMC reports a calibration figure of merit (FOM) for the calibration process. The factory calibrating procedure guarantees that DMC heading readings have the required and specified accuracy. The factory calibration procedure does not provide correction of the errors due to external factors (hard and soft iron errors); hence, additional compensation procedures should be developed.

The DMC is designed to provide digital communication with the host system. DMC could report the measured heading angle value, pitch and roll angles values, and all three magnetometer digital reading values. The factory calibration procedure and application of the FOM factor are extremely useful during the development phase when the DMC position is determined. It is also good to repeat it every time when system is initiated.

### 2.3. Digital Magnetic Compass—DMC Application Environment

DMC uses the Earth’s magnetic field properties to determine direction. The Earth’s magnetic field is influenced by molten iron in the Earth’s core and the Earth’s rotation; therefore, it is not always constant. This means that the local magnetic field varies in intensity and direction over the surface of the Earth [8,12]. The knowledge of the Earth’s magnetic field properties and anomalies is important for accurate direction finding.

In addition, magnetometer measurements have a lot of disturbances that contributes to DMC heading error. The accuracy of magnetometer measurements is reviewed [25], mathematically described [26], and compensated using developed calibration procedures [27] in the case of the three-dimensional aerospace magnetometers. These procedures and theory are partially applied in the commercial DMC based on solid-state magnetometers during compass design and development. The DMC user does not need to take direct care but should consider them in design phase to optimize DMC integration in the system.

Magnetometer errors should be considered during DMC design to minimize their influence on DMC heading accuracy. Furthermore, the magnetometer errors should be considered when magnetometers are used in a digital magnetic compass for heading measurements. Magnetometer error compensation in the case of the application for heading angle determination could be determined using slightly simplified theory as described in literature [28,29,30]. Magnetometer errors could be classified into several groups:

First of all, error sources are related to magnetometer sensor scaling, wideband noise, and temperature stability. Magnetometer signal A/D conversion should provide a suitable digitalization step for the required accuracy. Magnetometer scaling equalization could be improved through the calibration procedure.

The second group of error sources is related to non-orthogonality of the magnetometers and non-alignment with the body frame:

*Non-Orthogonality*: If sensor axes are not mounted orthogonally to one other, they will measure only a part of the magnetic field causing measurement errors depending on the non-orthogonality.

*Scaling:* There can be multiplicative errors in the magnetometer’s readings for the same magnetic field value in different measurement axes.

*Non-alignment:* During the system bore sighting procedure, non-alignment errors should be minimized or measured to be used as additive constants for automatic measurement error corrections.

The third group of error sources is related to the Earth’s magnetic field non-uniformities (disturbances) caused by soft-iron and hard-iron effects.

*Hard iron*: These disturbances are caused by the presence of ferromagnetic elements in the body reference frame and nearby measurement instruments. Hard-iron effects influence magnetometers and actually cause different offsets to the magnetometer reference axis.

Hard-iron distortion is produced by effects that generate a constant, additive field to the Earth’s magnetic field vector (permanent magnets (ruminant magnetization) and electrical currents (induced magnetization)), thereby generating a related additive magnetic field vector to the Earth’s magnetic field, causing change of the output signal of the each magnetometer axis sensors. A speaker magnet, for example, will produce a hard-iron distortion. The electrical current running through circuits near the magnetometer generates a magnetic field and will also introduce hard iron effects. As long as the orientation and position of the magnet relative to the sensor is constant, the disturbance magnetic field vector and associated offsets will also be constant.

*Soft iron:* Soft-iron effects are not directly caused by a magnetic field. Ferromagnetic materials on the platform interact with the outer magnetic field (the Earth’s magnetic field) disturbing it with differing strength, depending on the orientation of the material.

Unlike hard-iron distortion, where the generated magnetic field effect is additive to the Earth’s field as a vector, soft-iron distortion is the result of a material that influences, or distorts, a magnetic field but does not necessarily generate a magnetic field itself and is therefore not additive. Iron and nickel, for example, will generate soft-iron distortion. While hard-iron distortion is constant regardless of orientation, the distortion produced by soft-iron materials is dependent upon the orientation of the material relative to the sensor and the magnetic field. Thus, soft-iron distortion cannot be compensated with a simple constant; instead, a more complicated procedure is required.

A generalized measurement equation that involves all mentioned error influences is presented in Equation (3) [28]:(3)$$\vec{B^{h}}=C_{w}^{h}\cdot \left[C_{m}\cdot C_{sf}\cdot C_{si}\cdot \left(\vec{B^{w}}+\vec{H^{w}}+\vec{W^{w}}\right)\right]$$
where:
-$\vec{B^{h}}={\left[\begin{array}{ccc} B_{x}^{h} & B_{y}^{h} & B_{z}^{h} \end{array}\right]}^{T}$, magnetic vector components in direction measurement plane (horizontal)-$\vec{B^{w}}={\left[\begin{array}{ccc} B_{x}^{w} & B_{y}^{w} & B_{z}^{w} \end{array}\right]}^{T}$, magnetic vector components against DMC reference plane-$\vec{H^{w}}={\left[\begin{array}{ccc} H_{x}^{w} & H_{y}^{w} & H_{z}^{w} \end{array}\right]}^{T}$, hard iron bias influence on measurement-$\vec{W^{w}}={\left[\begin{array}{ccc} W_{x}^{w} & W_{y}^{w} & W_{z}^{w} \end{array}\right]}^{T}$, wide band noise influence on measured value-$C_{m},\;C_{sf},\;C_{si}$ are 3 × 3 matrices that account for soft iron, scale factor, and misalignment errors respectively-$C_{w}^{h}$ is a 3 × 3 matrix that represents coordinate transformation from the DMC body to the measurement plane

All influences as expressed in Equation (3) lead to compass deviation errors in a complex way and need to be compensated for to provide as accurate as possible heading measurements. The compass deviation compensation technique highly depends on the application. It is shown using Equation (3) that compensation could be designed as a combination of harmonic components, depending on the reported heading angle.

All these influences are listed to show how DMC application could be sensitive. The majority of errors are compensated during the DMC factory-defined calibration procedure, but additional DMC compensation of heading angle errors due to hard and soft iron influences in the field should to be developed.

### 2.4. Magnetic Declination

Assuming that a DMC is well designed, carefully manufactured, and calibrated, it still fails to determine the true (geographic) north due to two factors:

1. Magnetic deviation—the angle between the compass direction finding and magnetic north due to the presence of local disturbances of the Earth’s magnetic field. This could be corrected using a proper DMC heading compensation procedure.

2. Magnetic variation (or magnetic declination)—the angle between magnetic north and true north due to the local direction of the Earth’s magnetic field.

The Earth’s magnetic field variation during time and along the Earth’s surface has been studied for centuries to provide a proper data base for the necessary corrections.

It is known that the locations of the Earth’s magnetic poles are not coincident with the geographic poles and they are always wandering around. The Earth’s magnetic field is not a simple dipole, and geological masses can affect the local magnetic field. Furthermore, the Earth’s magnetic poles have the opposite orientation to the geographic poles. There are variations that change in time: slower ones due to changes in the Earth’s magnetic field, and more sudden and temporary ones due to sunspot activity and magnetic storms in the ionosphere.

The international geomagnetic reference field (IGRF) [31] is a series of mathematical models describing the large-scale internal part of the Earth’s magnetic field between 1900 A.D. and the present. The IGRF has been maintained and produced by an international team of scientists under the auspices of the International Association of Geomagnetism and Aeronomy (IAGA). This model provides new internationally created and available data every five years, starting from 1970, through model generations. Now the 12th generation is valid [31]. It uses the data sets of geomagnetic measurement results generated at dedicated measurement stations distributed over the Earth to calculate constants to be applied in the IGRF model for the determination of the local declination angle value. Examples of the values for the Earth’s magnetic declination and inclination values are presented in Figure 3 [31] as an illustration. The local magnetic declination values are usually incorporated into digital maps. In our application we are using that data. Local magnetic declination values incorporated in the digital map could be different from actual ones due to unexpected local influences. It is important to know that because an incorrect local magnetic declination value for some location could be an additional source of error for the measured azimuth angle. In our application, it is considered that a digital map contains correct declination angle values and it is used as is. In the case of errors in heading measurements, it is important to additionally consider local declination angle errors as a potential error source.

Once the proper compass and system bore sighting procedure is finished and the DMC calibration and compensation procedure applied, obtained heading angle values should be corrected using additive constants related to the measured misalignment error and declination angle to correct geographic (true) heading applicable for target position calculation.

### 2.5. Magnetic Compass Deviation Compensation

For centuries, a compass was a basic navigation tool for ships, but with the appearance of iron ships, the problem of compass deviation due to local disturbances has become even more important and must be treated seriously. In the mid nineteenth century, the first theories and compensation formulas and procedures appeared [32,33,34]. With time, the basic procedures for correcting ship magnetic compass deviation were refined and updated. All procedures were based on the measurement of the deviation of the selected direction and the prediction of the deviation in all other directions. In the case of a ship magnetic compass, adding of selected “magnetic” devices capable to minimize local disturbances was applied.

With the appearance of digital magnetometers, some modified procedures have arisen [15]. Wide application of magnetometers for navigation required new and real-time deviation compensation techniques and strategies [35,36,37,38,39,40], including Kalman filter-based methods, neural networks, non-linear approach, and fusion with gyro and inertial sensors.

DMC internal errors could be minimized through DMC calibration procedures that could be provided by the manufacturer and built-in DMC firmware. Usually, the DMC manufacturer provides a calibration procedure to be executed in the field that contributes to better accuracy in the field conditions.

External effects that cause DMC heading measurement errors could be compensated for by using the DMC compensation procedure.

The best-known and often applied procedure is measurement unit programed swing [32,33,34,35]. The leveled DMC is rotated by a full circle around the vertical axis (see Figure 4a), and related magnetometer and/or compass heading readings are recorded together with accurate swing angle values. The recorded measurements are used for DMC heading error compensation in the further DMC application.

There are two basic techniques for heading reading compensation and error minimization.

The first one is based on the application of magnetic field component measurements and heading angle calculation using compensated values.

The second is based on heading error modeling using the proposed error function depending on the measured heading angle and reading correction for the calculated (or tabulated) error value.


**Compass Swinging Procedure**


The illustration of the swinging procedure is shown in Figure 4a.

Following detailed theoretical heading error analysis [28], it was shown that the heading error (δψ) could be represented by Equation (4), following the detailed solution of Equation (3).
(4)δψ=Asinψ+Bcosψ+Csin2ψ+Dcos2ψ+E

Means, that the total heading deviation curve containing harmonic components are shown in Figure 4b.

Coefficients A, B, C, D, and E could be determined after measurements during the swinging process and represent disturbance influences as per Table 2 [8]. Some typical values for a ship compass are also listed in Table 2 just for the perception of the order of magnitude, but real values depend on the compass’ local magnetic environment.

The application of the swinging procedure is not always suitable because of two main shortcomings:

First, coefficients A–E are location dependent, so that the swinging procedure is not suitable for application on moving platforms, because multiple swinging should be applied during travel due to variations in the Earth’s magnetic field and localized hard and soft iron effects.

The second shortcoming of compass swinging becomes apparent when we note that the heading is a required input to the algorithm. Since heading errors due to hard and soft iron errors are heading dependent, the heading input into the algorithm will be corrupted by a non-constant bias. Thus, an independent measurement of the DMC position angle is required when calibrating magnetometers using this method.

In the case of a ship, it is applicable because the main disturbance is caused by the ship structure, so that once the ship is sailing on the open sea, there are no additional disturbances expected. In the case of a ship compass, the hardware solution for compensation is often successfully applied.

In the case of a stationary platform, the change of location is occasional without a change of position during system operation so the swinging procedure can be applied at the start and results could be used during the platform mission.

Multi-sensor surveillance systems having a precise pan/tilt platform provide reference angular measurements during swinging. It is also possible to determine heading after the DMC compensation procedure is applied and accompany it with related pan/tilt measurement results to later use the pan/tilt angle measurements as heading data as a redundant measurement channel because the pan/tilt platform position angle is usually measured with better accuracy than the DMC heading angle.

In the case of particular DMC heading reading errors being obtained during swinging steps (1, 2 … N), the measured errors should satisfy Equation (5). Solving Equation (5), a system of N linear equations for coefficients A, B, C, D, and E, provides DMC error compensation using Equitation (4). There is an optimal solution (see Appendix A) providing their values following minimal Euclidian distance criteria. That allows us to use coefficients A, B, C, D, and E for the correction of any DMC heading reading with good accuracy during the stationary application of the multi sensor system.
(5)$$\left[\begin{array}{c} \begin{array}{c} {\delta \psi }_{1}\\ {\delta \psi }_{2}\\ \vdots \end{array}\\ \vdots \\ {\delta \psi }_{n} \end{array}\right]=\left[\begin{array}{c} {\sin\psi }_{1}\\ \sin\psi_{2}\\ \begin{array}{c} \vdots \\ \vdots \\ \sin\psi_{n} \end{array} \end{array}\begin{array}{c} {\cos\psi }_{1}\\ {\cos\psi }_{2}\\ \begin{array}{c} \vdots \\ \vdots \\ {\cos\psi }_{n} \end{array} \end{array}\begin{array}{c} \sin2\psi_{1}\\ \sin2\psi_{2}\\ \begin{array}{c} \vdots \\ \vdots \\ \sin2\psi_{n} \end{array} \end{array}\begin{array}{c} \cos2\psi_{1}\\ \cos2\psi_{2}\\ \begin{array}{c} \vdots \\ \vdots \\ \cos2\psi_{n} \end{array} \end{array}\begin{array}{c} 1\\ 1\\ \begin{array}{c} \vdots \\ \vdots \\ 1 \end{array} \end{array}\right]\cdot \left[\begin{array}{c} A\\ B\\ \begin{array}{c} C\\ D\\ E \end{array} \end{array}\right],$$

Once we decided to use the swinging procedure for DMC compensation, the next step was to select the number of the swinging steps. Two options appeared as the logical solution:

**Simplified compensation**: Use basic directions (**N**—north (0°), **NE**—north-east (45°), **E**—east (90°), **SE**—south-east (135°), **S**—south (180°), **SW**—south-west (225°), **W**—west (270°), and **NW**—north-west (315°) as usual for ship compass compensation. These basic directions are illustrated in Figure 3 and annotated as measurement points 0–7, respectively.

If we use simplified swinging, it is possible to define the solution for compensation coefficients that use all eight measurements (as listed in Equation (6)):(6)$$E=\frac{\Delta \psi_{0}+\Delta \psi_{2}+\Delta \psi_{4}+\Delta \psi_{6}}{4}\;A=\frac{\Delta \psi_{1}+\Delta \psi_{3}-2\cdot E}{\sqrt{2}}\;B=\frac{\Delta \psi_{1}+\Delta \psi_{7}-2\cdot E}{\sqrt{2}}\;C=\frac{\Delta \psi_{1}+\Delta \psi_{5}-2\cdot E}{2}\;D=\frac{\Delta \psi_{0}+\Delta \psi_{4}-2\cdot E}{2}$$

**Full compensation**: Use measurements every 10°—36 points—which requires more time but delivers better accuracy. For the case of full compensation, we used an optimal algorithm described in Appendix A to calculate compensation coefficients.

### 2.6. Digital Magnetic Compass Alignment with Optical Axis

In the electro-optical multi-sensor systems, a DMC sensor is used for north finding (heading angle) measurements as a base functionality used for system orientation. In addition, DMC pitch and roll measurement results could be used when selected target coordinates are calculated [24] using a related measurement equation. To provide EO multi-sensor system capability for target coordinates determination with sufficient accuracy, the proper bore sighting procedure should be applied during system integration. Electro-optical sensors (all imagers and laser range finders) are aligned using a collimator. It is possible to provide that the collimator optical axis is set in the horizontal plane so that it could be used as a reference for the DMC body frame definition. This reference could be transferred to the EO system position using an accurate device under test (DUT) manipulator and pan/tilt (P/T) platform position control and measurements. By using DUT manipulator and P/T commands, the EO multi-sensor optical axis could be positioned in the horizontal plane.

The DMC pitch and roll angle readings recorded during the bore sighting procedure while the system optical axis is in the horizontal plane could be used as additive constants for corrections of the real readings obtained during the field operation.

EO system optical axis and heading angle misalignment could be corrected using an additive constant, but to provide measurements, one needs to provide a proper known reference that could be detected by EO sensor and DMC at the same time. This is not easy to provide in the laboratory. In the field condition, using a known target with known coordinates, it is possible to determine the optical axis and the DMC heading misalignment angle value.

### 2.7. Digital Magnetic Compass Integration Issues

The DMC contains solid-state magnetometers, inertial sensors, and a processor that can be accessed or programmed using an interface serial port allowing the designer to set the measurement frequency, integration time, and format of the output data. In addition, there is a DMC development kit that provides the proper DMC setting and calibration. The DMC has the capability to transmit “raw” data (magnetic field components (x, y, z) in digital counts values, the calculated azimuth angle, and measured roll and pitch angles at a selected frequency and data integration time. DMC precision is therefore influenced by the transmission frequency and A/D conversion steps. The DMC housing has a defined connection plane and could be built to provide different line of sight orientation.

The EO multi-sensor system contains many electronic boards and electromechanical components that are able to disturb magnetic readings. The first step is to find the proper place for the DMC mounting providing good DMC line of sight and EO system optical axis alignment and the lowest disturbance of magnetometers. Otherwise, the DMC should be mounted on the outer bar, far away from the EO sensor housing, as in [41]. The proper DMC mounting position should provide that the selected factory calibration procedure can be executed with satisfactory FOM.

The second step is to define the measurement frequency and integration time that should provide good measurement precision and measurement, calibration, and compensation procedure execution in the time frame that is allowed by the EO system mission requirement (inside system readiness time).

The third step is to select the DMC heading deviation compensation procedure that is easy for execution and to use the benefits of the EO system architecture and components capabilities (for example, pan/tilt motion speed range and position precision measurements).

The fact that the system is used in stationary position and has a high-precision pan/tilt platform allows that the DMC’s role can be reduced to only the system line of sight orientation task at the start, when the system line of sight could be placed in or close to the horizontal plane. In that case, one can use the DMC heading error compensation procedure, providing high heading accuracy at the same time.

The pan/tilt platform is used as an angular measurements reference platform [42] during the swinging procedure. The quality of the factory calibration provides that external magnetic disturbances could be compensated using a traditional compensation equation having harmonic components.

The manufacturer’s DMC calibration procedure also requires programmed motion and measurement execution at predefined positions, and it is performed as a first step for DMC setting.

DMC deviation compensation (hard iron and soft iron) (as illustrated in Figure 5) are performed as a second step. After that, the EO multi-sensor system is correctly oriented and can be used for target position determination.

The EO system heading angle can be measured using compass heading angle readings or, alternatively, pan/tilt position angle readings.

The technologies used in EO multi-sensor surveillance systems (optical, GNSS, mechanical, laser range finding, magnetic, and inertial) are suitable for target recognition and position estimation [6,41], but proper sensor data fusion should be applied [43].

Using the system operator’s console, the system geolocation data could be presented to be continuously tracked. Target position data could be integrated in the digital map used in the system or transferred over network.

## 3. Measurement Results

During the development phase, we performed testing with special attention on DMC deviation to provide compensation procedures suitable for achieving required heading reading accuracy.

DMC deviation angle was determined using pan/tilt angle measurements as a reference. Due to the harmonic nature of the deviation, it is possible to find pan/tilt-to-DMC reading off-set values.

Testing results show that DMC deviation angle measured values during the swinging procedure are sensitive to outer conditions and the system’s position, as expected. It has been confirmed once again that a compensation procedure is necessary. In Figure 6, we presented the worst-case results (CASE 1) obtained during testing containing unknown measurement irregularity in addition to soft- and hard-iron effects. In Figure 7, we present selected measurement results for one of typical measurement conditions without irregularities (CASE 2). We applied compensation procedures (simplified and full) for both measurement data sets and calculated the residual error.

The residual error distribution is presented in Figure 8, and it was the same for both compensation procedures (full and simplified). In Table 3, the statistical parameters of the residual error distribution are presented. Although the residual error distribution graph is the same for both compensation procedures, the differences in the statistical parameters (mean value and standard deviation) show that full compensation is able to provide better accuracy.

Field trials for heading angle measurement accuracy were performed using several selected targets with accurately known target’s referent coordinates. The analysis of measurement results provides us with DMC and optical axis misalignment angle values used for DMC reading correction. After additive correction applied on field heading angle measurement error determination, we found that for a set of three predefined targets and two measurements for each target, the residual measurement error was in the range of −0.15−+0,15 degrees, which was better than the required value for the target position CEP50 error.

## 4. Discussion

Pan/tilt angular position during swinging procedure execution is considered as reference because angle measurement accuracy is 0.1°, which is better than the expected DMC heading angle accuracy (0.25° (1σ)). The pan/tilt position angle has an off-set related to the DMC heading angle defined as the difference between DMC heading reading and pan/tilt position reading. Due to DMC reading deviation this off-set is not constant. Because DMC heading reading deviation has an expected harmonic nature (see Figure 3b), the mean value of such defined off-set is a real constant off-set. The pan/tilt reading corrected with the real constant off-set is now the reference azimuth angle for the DMC heading. The difference between the reference azimuth angle and the DMC heading reading is equal to the DMC reading deviation. The DMC deviation angle and the DMC heading reading are used to calculate the compensation coefficients and DMC reading compensation.

Residual error is defined as the difference between the calculated DMC compensation angle and DMC deviation angle. The residual error distribution shows the quality of compensation in the case that irregular disturbance during the swinging generated measurement error could not be fully compensated. In the case of regular measurements without unexpected disturbances, the error distribution has the expected normal distribution.

The applied compensation procedures (full, 36 points; simplified, 8 points) behave in a similar way regarding error distribution, but the full compensation procedure provides better accuracy (lower mean value and standard deviation), as shown in Table 3.

According to initial testing results using the swinging procedure, the full compensation procedure is adopted as the final solution and incorporated into the system control software following the procedure flow chart presented in Figure 5.

The factory DMC calibration procedure and swinging-based soft- and hard-iron compensation procedure execution time is less than the system’s required readiness time. In the case that the time available for the DMC compensation procedure is short, the simplified compensation procedure could be applied, but some loss of accuracy should be expected (see Table 3).

In the case that the calibration and compensation procedures fail, the EO system software provides that the system can be applied normally using manually introduced orientation data during the system’s initiation procedure.

## 5. Conclusions

The basic principles of the Earth’s magnetic field measurements and their application for north finding (heading angle measurements) are reviewed to identify key measurement error sources in the DMC application for multi-sensor electro-optical system orientation.

A concise overview of the complex EO multi-sensor system structure and architecture having geo-referencing capability of a selected target is presented.

Systems engineering principles are applied, providing simplification of the DMC calibration and compensation taking care of the other system components capabilities and system mission characteristics. This approach provides that applied technical solutions could be simplified whilst fitting other system requirements and providing high target geo-referencing accuracy.

Sensor integration and data fusion should provide the required accuracy for target location estimation and presentation on the digital map.

The magnetometer error sources that could have an influence on DMC heading angle measurements are described to provide a better understanding of DMC application environment influences on the DMC errors. The majority of these errors are corrected using the factory proprietary calibration procedure, which is not described in this article.

The DMC magnetic deviation compensation procedure that uses the manufacturer’s calibration and additional external disturbances compensation (soft- and hard-iron effects) using traditional compass swinging supported with pan/tilt precise position angle measurements was developed and practically implemented in the system which is now in the exploitation.

The practical engineering solution for DMC application in EO system orientation, using system architecture, mission, and components properties to simplify DMC calibration and compensation, was realized.

The DMC design-related measurement errors were corrected automatically using the factory DMC calibration procedure in the application environment. The measurement errors due to the DMC and optical axis misalignment were corrected using a system bore sighting procedure in which additive compensation constants were determined. Soft and hard iron caused measurement errors are minimized using the swinging-based compensation procedure.

We described the swinging procedure applied and presented experimental results obtained during initial DMC design examination. Our experimental results show that the full compensation procedure provides slightly better heading angle measurement accuracy but could not compensate for accidental measurement disturbances. The compensation procedure is based on well-known principles, but the selected technical solution and practical realization provides reliable and sufficient heading angle measurement accuracy which does not disturb system operation.

The application of the DMC used in the multi-sensor electro-optical system used for system geo-referencing functionality is described. The achieved heading angle measurement accuracy fulfills the requirements. It is shown that application of the DMC in a multi-sensor system is able to provide system geo-referencing functionality, but application of the DMC calibration, misalignment correction, and soft- and hard-iron compensation should be carefully implemented.

The results presented it this paper show that a complex sensor could be successfully integrated into an EO multi-sensor system, using a relatively simplified technical solution based on a good understanding of the component capabilities and providing satisfactory accuracy for defined system mission.

## Figures and Tables

**Figure 1 sensors-19-04331-f001:**
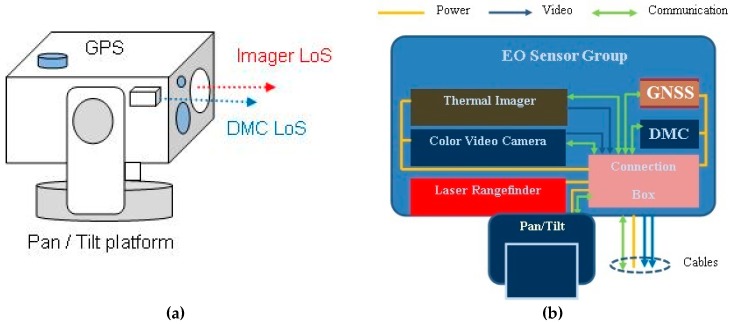
Long-range electro-optical multi-sensor structure: (**a**) general view of sensor mounting, (**b**) functional block diagram of the electro-optical head.

**Figure 2 sensors-19-04331-f002:**
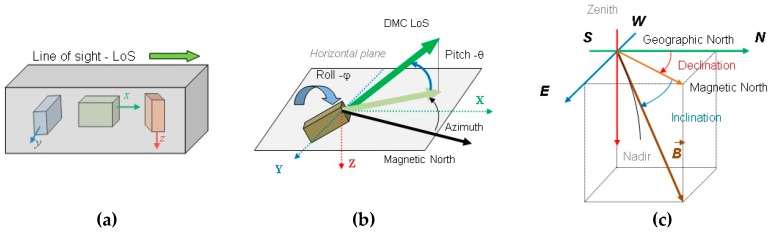
Digital magnetic compass (DMC) orientation geometrical parameters: (**a**) DMC magnetometers internal orientation, (**b**) DMC orientation against horizontal plane, (**c**) measured magnetic north against geographical north.

**Figure 3 sensors-19-04331-f003:**
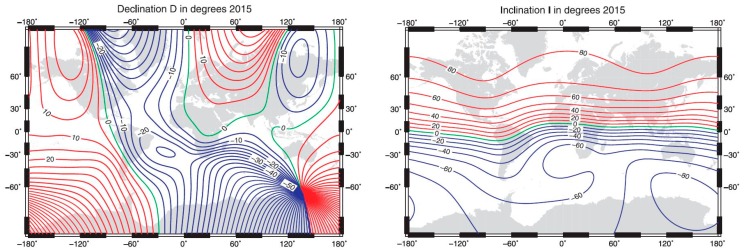
The Earth’s magnetic field model results for declination and inclination values, as per international geomagnetic reference field (IGRF) 12 GEN.

**Figure 4 sensors-19-04331-f004:**
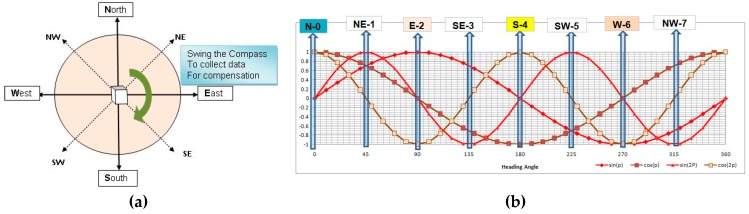
Swinging procedure: (**a**) graphical description of swinging, (**b**) compensation harmonic components.

**Figure 5 sensors-19-04331-f005:**
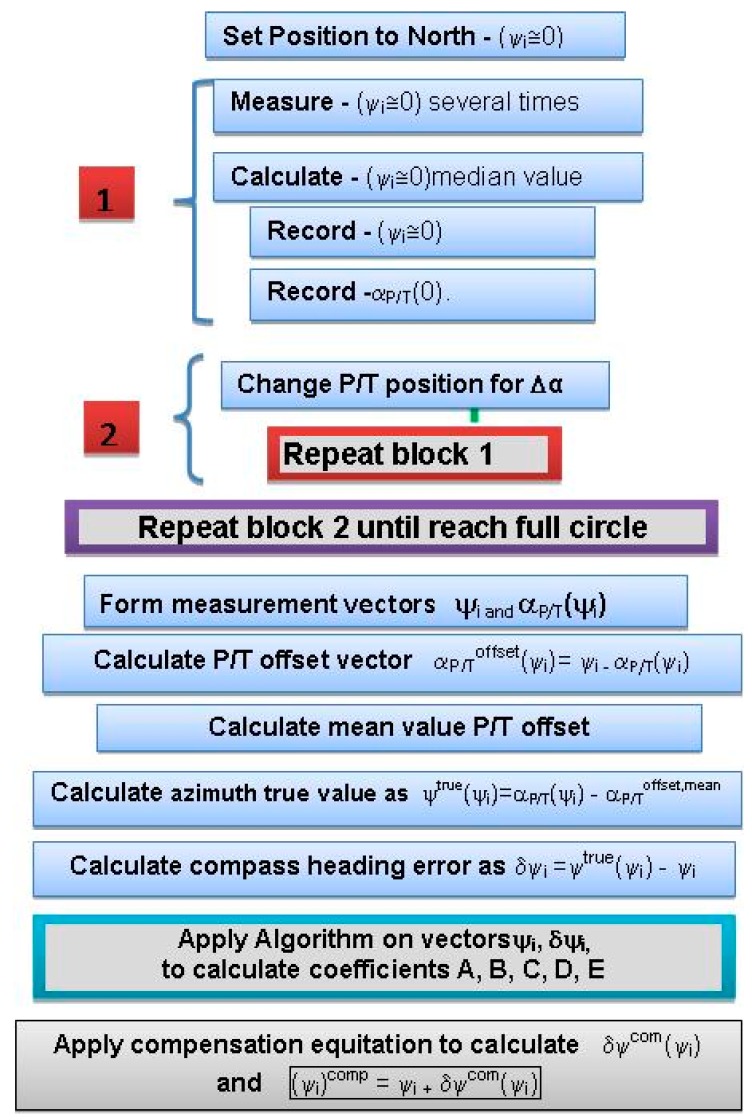
Compass deviation compensation procedure flow chart.

**Figure 6 sensors-19-04331-f006:**
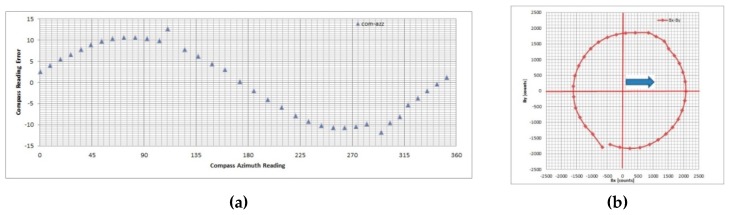
Compass deviation measurement results (CASE 1) with excess irregularity during measurements: (**a**) compass deviation angle vs. compass reading, (**b**) magnetic field projection in horizontal plane.

**Figure 7 sensors-19-04331-f007:**
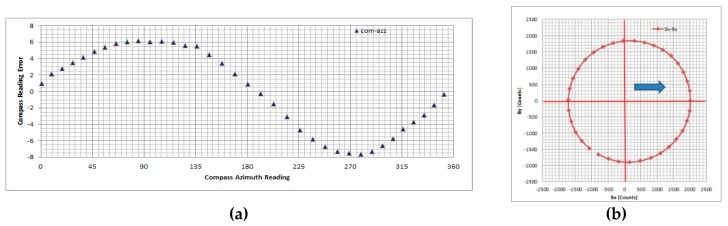
Compass deviation measurement results (CASE 2) without irregularity during measurements (regular): (**a**) compass deviation angle vs. compass reading, (**b**) magnetic field projection in horizontal plane.

**Figure 8 sensors-19-04331-f008:**
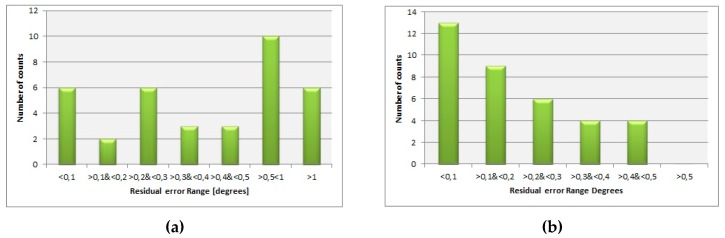
Residual error distribution after compensation: (**a**) CASE 1, with accidental irregularity during measurements; (**b**) CASE 2, normal operation during measurements.

**Table 1 sensors-19-04331-t001:** DMC basic parameters.

Characteristics	Value	Units
***Azimuth (heading) accuracy***	0.25	*Degrees (1 σ)*
***Elevation (pitch)/bank (roll) accuracy***	±0.1	*Degrees*
***Weight***	25	*grams*
***Operating temperature range***	−32–+55	°C

**Table 2 sensors-19-04331-t002:** Compensation coefficient type and error assigned.

Coefficient	Deviation Type	Error Causes	Typical Value
***A***	Semicircular, *sinψ*	Induced hard iron in vertical plane in front or back of compass	**~10°**
***B***	Semicircular, *cosψ*	Induced hard iron in vertical plane on the side of compass	**~6°**
***C***	Quadrantal, *sin2ψ*	Induced magnetism in all symmetrical arrangements of horizontal soft iron	**~4°**
***D***	Quadrantal, *cos2ψ*	Induced magnetism in all asymmetrical arrangements of horizontal soft iron	**~1.5°**
***E***	Constant	Physical: Magnetometer and compass non-alignment Magnetic: Asymmetric influence of soft-iron disturbance	**~1°**

**Table 3 sensors-19-04331-t003:** Residual error distribution statistical parameters.

Residual Error Statistical Parameters [Degrees]	CASE 1 (Irregular)		CASE 2 (Normal)	
	Full 36 Points	Simplified 8 Points	Full 36 Points	Simplified 8 Points
***Average value***	0.0667	0.0279	0.00041	–0.00825
***Standard Deviation***	0.44764	0.5679	0.11174	0.24604

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
