# Peer review of "Digital Magnetic Compass Integration with Stationary, Land-Based Electro-Optical Multi-Sensor Surveillance System"

_sensors, 2019, doi:10.3390/s19194331_

Round 1

Reviewer 1 Report

The authors have made a careful analysis of my comments. Generally, they follow my recommendations, but stay unclear for some of them (heading computation, Eq. 1-2; magnetometer compensation). I understand that these are not key topics of the manuscript. This is why I recommend now publication of this work.

Line 581. Typo Leach and not Lich

Author Response

The authors have made a careful analysis of my comments.

Once again, thank you for your detail comments in the previous review.

We are always considering reviewer’s comments seriously: (1) Reviewers are selected as experts in the field so the level of expertise is significant, even it can be centered narrow against the topic treated in article, so their opinion is very important. (2) Reviewers had more experience so their capability help is valuable. (3) In the case when reviewer identifies something as not clear enough there is a chance that common reader could be confused. Means reviewer’s opinion should be highly respected.

Generally, they follow my recommendations, but stay unclear for some of them (heading computation, Eq. 1-2; magnetometer compensation). I understand that these are not key topics of the manuscript. This is why I recommend now publication of this work.

We are using commercial DMC, capable to deliver heading data with required accuracy after manufacturer’s calibration procedure is executed. We are using that procedure but manufacturer do not disclose the all technical details or theoretical background used for procedure development.

We use this simplified theory just to illustrate that calibration process is complicated due to number of influences.

In the practical application it shows up that manufacture’s calibration procedure do not correct all magnetometer errors. To distinguish the errors introduced from outside effects (so called hard and soft iron) we introduced additional compensation procedure.

We reviewed once again introduction and other chapters to make changes that these facts are more clearly pointed out.

Reviewer 2 Report

This paper presents a calibration procedure to calibrate digital magnetic compass. The authors have introduced many error sources that will affect the DMC measurement accuracy. The following points need to be addressed:

I cannot find detail process to obtain the unknown variable introduced in section 2.3. How to estimate them? The experimental results are lacking. It would be nice if the improvement can be seen after each step. For the moment only Figure 7 presents the calibration performance, which is far from sufficient.  Have you considered the practical use of proposed method? Once the calibrated sensor (DMC) is placed/integrated in the system, many errors are again introduced. In such condition, processes such as pointing to north or swinging are almost impossible to be done by the system integrators or customers. What would be the solution for such scenario? 

Author Response

This paper presents a calibration procedure to calibrate digital magnetic compass.

This paper treats the integration of DMC with complex EO system.

The DMC is component from the market that should be calibrated once when it is built in using manufacturer’s proprietary procedure but additional compensation procedure is required for soft and hard iron influence compensation

The authors have introduced many error sources that will affect the DMC measurement accuracy.

Error sources are introduced only to illustrate how complicated is DMC as component and to show how complicated is to integrate it with EO system. DMC calibration procedure is manufacturer’s proprietary procedure that could not be disclosed or not known in details. After calibration is done additional compensation procedure we described in the paper should be performed.

I cannot find detail process to obtain the unknown variable introduced in section 2.3. How to estimate them?

The whole section 2.3 is written only to illustrate the DMC complexity and application issues even one use the best and mature component. Manufacturer’s proprietary calibration procedure is applied to get DMC heading readings with required accuracy.

The experimental results are lacking. It would be nice if the improvement can be seen after each step. For the moment only Figure 7 presents the calibration performance, which is far from sufficient.

We selected only representative experimental results that lead us to selection of the type of swinging (8 or 36 points). We did lot of experiments but we presented only the worst case and normal case, which is sufficient to show the difference between these two types of swinging.

Figure 6 shows the results of the heading accuracy when only manufacturer’s calibration procedure is executed and why additional compensation is necessary.

Figure 7 and Table 3 together show the differences. The practical field trials results also show over all calibration and compensation procedure performance.

Have you considered the practical use of proposed method?

As system integrator we implemented proposed method in our system. The key contribution of our article is practical implementation of the DMC in the system

Once the calibrated sensor (DMC) is placed/integrated in the system, many errors are again introduced. In such condition, processes such as pointing to north or swinging are almost impossible to be done by the system integrators or customers. What would be the solution for such scenario?

DMC application in the EO multi-sensor system is aimed to support remote object of interest (target) coordinate determination using GNSS data, LRF and digital map. DMC is used to determine North position (EO system orientation) and object azimuth angle.

The described method consisting of DMC calibration using manufacturer’s calibration procedure that provides provide DMC heading reading with required accuracy and additional compensation procedure correcting DMC errors (soft and hard iron) not covered by calibration are applied every time when EO system is activated  using time slot (system readiness time) during thermal imager cool down process.

We are system integrator and we incorporated calibration and compensation procedure using systems architecture.

In the case when DMC is not activated properly (calibration and compensation procedure are not executed properly) there is reserve solution for manual determination of the north direction and connection with digital map.

We reviewed once again introduction and other chapters to make changes pointed out more clearly what was done:

Introduction is changed to point out that practical engineering solution for DMC integration is not easy task but could be resolved using EO system architecture specifics. Section 2.2 is changed to explain difference between calibrations and follow on compensation procedures. Sections 2.3 and 2.7 are changed to clarify DMC integration and application in the system during system mission. Measurement results are related only to determine efficacy of swinging procedure and to provide selection of the swinging parameters. Although, a lot of measurements were done we selected to present only one irregular swing (the worst case) and selected experiment of all rest “regular” swing cases without accidental disturbance. So, experimental results section is reviewed, once again, but not significantly changed. The Section 4 is slightly changed to pint out more clearly that measurement result confirm that swinging is applicable and that selection of the swinging procedure applied is possible. Also, we provide more detail explanation what can be used in the case that calibration and compensation procedure fail and EO system could be applied normally using manually introduced orientation data. The Section 5 is slightly corrected to point out that we are integrator of the whole system and we developed additional procedure to compensate heading reading as a part of the DMC integration process. During the revision process we also corrected some mistyping, spelling and grammar errors.

Round 2

Reviewer 2 Report

Most of the points have been addressed. 

This manuscript is a resubmission of an earlier submission. The following is a list of the peer review reports and author responses from that submission.

Round 1

Reviewer 1 Report

The manuscript submitted by Livada et al. entitled "Digital magnetic compass integration with stationary, land based electro-optical multi-sensory surveillance system”» presents the procedures to use a digital magnetic compass in a global navigation satellite system. The main goal of the manuscript is to describe the computations and procedures to correct the defaults of the magnetometer and its magnetic environment to obtain accurate heading values. Some measurements and results are also shown to establish the heading compensation procedure.

As a geophysicist, I am not a specialist of multi-sensor imaging systems and I cannot attest the originality of this work. However, I am not sure of the novelty of this work and the part concerning magnetic measurements and corrections is poorly addressed. The question of the accuracy of three component magnetic measurements, magnetometer error corrections and magnetic compensation is well known from satellite and airborne (aircrafts or drones) measurements and there are tens (perhaps hundreds) of papers discussing the theory and applications of these questions. What is discussed in sections 2.2 and 2.3 is unclear, not original to my opinion, and shows confusions.

1)       Equation 2 should not be valid if pitch or roll is near +/-90. On a general point view, equations should be expressed in 3D after having defined the magnetometer reference frame and the geographical reference frame. This is what is done in the literature.

2)      Three component magnetometer errors are not well described. See, for example Olsen et al. (2001) who give a good review of the error computations and propose a software to compute them. Line 126-127, the assertion is wrong (“if there is an orthogonality error, the magnetometer will measure only a part of the magnetic field), but magnetic measurements will be in error, more or less important, depending the orthogonality errors (e.g. Olsen et al., 2001)

3)      Magnetic compensation is also not well explained. What is called hard iron or soft iron corresponds to magnetic fields due to remanent and induced magnetizations attached with the magnetometer (Leliak, 1961; Leach, 1980). The remanent magnetic field is constant in the magnetometer reference frame, not in the geographical reference frame. It the contrary for the induced magnetic field. Line 146-147, the assertion is non-sense: the remanent magnetic field is non additive to the Earth’s field. It can be the contrary, depending the relative orientation. Induced magnetization produces a magnetic field in all cases.

4)      Equation 3 is particularly unclear. From where is it coming looking what as explained before? See Leliak (1961) and Leach (1980) for a good mathematical description.

5)      Finally, to compensate, there are nine parameters to compute (Leach, 1980) not five. I understand that the authors are only interested by obtaining a “compensated” heading, but what happens if the pitch or the roll are not null? There is also the question of magnetic fields due to eddy currents (Leach, 1980). This possible problem should be at least mentioned.

6)      The question of magnetic declination is not well addressed. The IGRF is not a database, but coefficients computed using spherical harmonic analysis and data from magnetic observatories and satellites. The declination is computed from these coefficients and the corresponding software all over outside the surface of the Earth and at any moment. Magnetic declination can deviate a lot because of local magnetic field, in particular in urban environment but also, in some cases, in natural environments, in particular in areas where some kind of volcanic rocks are abundant.

Reviewer 2 Report

1 In this paper, a universal method is applied to build the system. Multi-sensor imaging systems using Global Navigation Satellite System (GNNS) and Digital Magnetic Compass (DMC) for geo-referencing are also widely used. For example, Zinner M, etc. mentioned “With the laser range finder, the Digital Magnetic Compass and GPS it is possible to measure one’s own and the target's position.”[1]

2 Compensation procedure is based on the well-known principles. The selected technical solution for practical realization provides reliable and sufficient heading angle measurement accuracy, but the compensation method of DMC error is only applied to different scenarios. Therefore, the innovation is insufficient in this paper.

3 This paper is not suitable for this journal. It is recommended to choose another journal.

4 The Global Navigation Satellite System is abbreviated as GNSS, not GNNS.

5 Figure 7 and Figure 8 need to be reconfirmed, because the graphics are exactly the same.

6 For the experimental conclusions, the experimental parameters, experimental procedures and experimental results description analysis are not clearly given.

[1] Zinner M , Krause U , Heinrich J . SPIE Proceedings [SPIE SPIE Defense, Security, and Sensing - Orlando, Florida, USA (Monday 13 April 2009)] Infrared Technology and Applications XXXV - Handheld multifunctional thermal imager and surveillance instrument from Jena-Optronik as part of the German \"IDZ-Infanterist der Zukunft\" project[J]. 2009, 7298:72981F.

Reviewer 3 Report

The manuscript describes the DMC compensation technique suitable for operation environment and discusses various technical solutions. Particularly, the swinging procedure is presented as a potential solution for DMC compensation in the given application. Both theoretical and experimental results are presented indicating possibility to integrate DMC in the long-range surveillance systems providing required geo-referencing data. The authors provide valuable knowledge on how to implement the compensation procedure and integrate the magnetic sensor.

The following references are missing in the text – omit them from the reference list (with necessary re-numbering of the rest) or add missing references to the text (easier):

9,10, 23, 36,37,38

Quality of the manuscript is impaired due to some grammar inconsistencies.  Please, make at least the following language corrections:

163- … that represents …

182 - … that contributes…

201- … curve contains…

203- …and represent …

205- …values only depend on…

208- …suitable because of two…

224- …provides…

231-…equations for… using equation (3).

235- …next step was…

286 – that… (Revise grammar of the following sentence)

302- … in Figure…

320- … provides us…

322-… set of three targets…

374- … defines…

393- … satisfy…

No other comments and/or questions.